# Silencing of *Glycogen Synthase Kinase 3* Significantly Inhibits Chitin and Fatty Acid Metabolism in Asian Citrus Psyllid, *Diaphorina citri*

**DOI:** 10.3390/ijms23179654

**Published:** 2022-08-25

**Authors:** Jin-Bo Zhang, Zhan-Jun Lu, Hai-Zhong Yu

**Affiliations:** 1College of Life Sciences, Gannan Normal University, Ganzhou 341000, China; 2National Navel Orange Engineering Research Center, Ganzhou 341000, China; 3Ganzhou Key Laboratory of Nanling Insect Biology, Ganzhou 341000, China

**Keywords:** chitin, DcGSK3, *Diaphorina citri*, fatty acid, RNA interference, transcriptome sequencing

## Abstract

Glycogen is a predominant carbohydrate reserve in various organisms, which provides energy for different life activities. Glycogen synthase kinase 3 (GSK3) is a central player that catalyzes glucose and converts it into glycogen. In this study, a *GSK3* gene was identified from the *D. citri* genome database and named *DcGSK3*. A reverse transcription quantitative PCR (RT-qPCR) analysis showed that *DcGSK3* was expressed at a high level in the head and egg. The silencing of *DcGSK3* by RNA interference (RNAi) led to a loss-of-function phenotype. In addition, *DcGSK3* knockdown decreased trehalase activity, glycogen, trehalose, glucose and free fatty acid content. Moreover, the expression levels of the genes associated with chitin and fatty acid synthesis were significantly downregulated after the silencing of *DcGSK3*. According to a comparative transcriptomics analysis, 991 differentially expressed genes (DEGs) were identified in ds*DcGSK3* groups compared with ds*GFP* groups. A KEGG enrichment analysis suggested that these DEGs were primarily involved in carbon and fatty acid metabolism. The clustering analysis of DEGs further confirmed that chitin and fatty acid metabolism-related DEGs were upregulated at 24 h and were downregulated at 48 h. Our results suggest that *DcGSK3* plays an important role in regulating the chitin and fatty acid metabolism of *D. citri*.

## 1. Introduction

Citrus Huanglongbing (HLB) is one of the most destructive citrus diseases worldwide, caused by the bacterium *Candidatus* Liberibacter asiaticus (*C*Las) and spread by *Diaphorina citri* Kuwayama [1,2]. The management of HLB can be achieved based on preventive measures, such as planting healthy nursery trees, eliminating *C*Las-infected trees and controlling the transmission vector [3]. In all the prevention and control measures of HLB, the most effective method is the vector control of *D. citri* [4]. To date, the control of *D. citri* mainly relies on the use of chemical insecticides, including neonicotinoids, pyrethroid and organophosphate [5]. However, the excessive use of chemical insecticides has led to high levels of insecticide resistance in *D. citri* [6]. Therefore, it is crucial to screen and investigate novel targets for controlling *D. citri*.

In most organisms, glucose metabolism plays a crucial role in energy metabolism, which provides the energy required for life processes, such as growth and development [7,8]. Glycogen and triglyceride are two major energy reserves in animal cells; however, their storage forms are different [9]. Glycogen is stored in a bulky hydrated form, whereas triglyceride is stored in an anhydrous form [10]. Glycogen is a branched glucose polymer that stores energy and carbon in organisms, which plays an essential role in maintaining glucose and energy homeostasis [11]. The glycogen in animals is synthesized under the catalysis of glycogen synthase, and glycogen degradation requires the action of glycogen phosphorylase, debranching enzymes and acid α-glucosidase [12,13]. However, in insects, glycogen is primarily synthesized from UDP-glucose and is used as spare sugar along with trehalose in order to maintain glucose availability [14]. Therefore, glycogen metabolism in insects is closely related to the formation and utilization of trehalose. Glycogen synthase kinase 3 (GSK3) is a serine–threonine kinase that acts as an important regulator of metabolism and development in diverse organisms [15]. Numerous studies have confirmed that GSK3 plays an essential role in regulating glucose metabolism [16,17,18]. GSK3 can inhibit glycogen synthase in insects by phosphorylating and blocking the transmission of the insulin signaling pathway [19]. Ding et al. revealed that GSK3 is a key enzyme that regulates energy and chitin metabolism in *N. lugens* [20]. However, the molecular mechanisms by which GSK3 regulates chitin and fatty acid metabolism in *D. citri* have not yet been reported.

RNA interference (RNAi) is a conserved mechanism in most organisms causing a sequence-specific gene silencing phenomenon induced by double-stranded RNA (dsRNA) [21]. RNAi has recently been developed as a useful tool for functional gene research in animals and plants [22]. The silencing of some target genes in insects by ingesting dsRNA suggests that RNAi technology may have application in pest control. Mao et al. revealed that silencing the *Helicoverpa armigera P450 monooxygenase* gene by plant-mediated RNAi impaired the larval tolerance of gossypol [23]. Therefore, RNAi has shown tremendous potential in pest management [24]. There are three approaches to deliver dsRNA to insects, including oral feeding, microinjection and topical applications, but the microinjection of dsRNA can only be used for fundamental research in the laboratory [25]. Furthermore, the transgenic plants expressing the dsRNA of specific genes are also receiving increasing attention [26]. However, RNAi efficiency is influenced by several external factors, including incomplete dsRNA internalization, instability of dsRNA, impaired systemic spreading of the RNAi signal and refractory target genes [27]. Hence, an efficient dsRNA delivery method and the selection of target genes are the two prerequisites for RNAi-mediated pest management. RNAi by the oral delivery of dsRNA in insects has great potential as a tool for integrated pest management [28]. Cao et al. demonstrated that silencing *Mythimna separate* chitinase genes by an oral delivery of dsRNA caused lethal phenotypes [29]. The silencing of *D. citri NADPH-cytochrome P450 reductase* (*DcCPR*) by oral delivery significantly decreased the expression level of *DcCPR* and increased the susceptibility of *D. citri* to chemical insecticides [30]. Moreover, the silencing of the *D. citri trehalase* (*DcTre*) gene by RNAi significantly reduced the expression levels of chitin-metabolism-related genes, led to a malformed phenotype and dsRNA delivery was conducted by an artificial diet [31]. In hemipterans, dsRNA is easily degraded by either the high pH or dsRNase in the gut [32]. Consequently, we conclude that the microinjection of dsRNA is more than that of oral administration.

In the present study, a *glycogen synthase kinase 3* (*GSK3*) gene, named *DcGSK3*, was identified from the *D. citri* genome database, and the sequences and spatial and temporal expression profiles of *DcGSK3* were analyzed. The functions of *DcGSK3* were investigated using RNAi combined with transcriptome sequencing. Furthermore, the levels of glucose, trehalose, glycogen, free fatty acids and trehalase activity were measured. The results showed that the silencing of the *DcGSK3* gene significantly inhibited chitin and fatty acid metabolism, indicating that *DcGSK3* is involved in regulating chitin and fatty acid synthesis. These findings provide a theoretical basis for *D. citri* control through chitin and fatty acid metabolism.

## 2. Results

### 2.1. Bioinformatics Analysis of DcGSK3

By blasting the glycogen synthase kinase 3 protein sequence from *N. lugens* (XP_039293499), *Bombyx mori* (XM_004928285) and *Drosophila melanogaster* (NP_476714) against the *D. citri* genome database in NCBI, the *DcGSK3* cDNA sequence of 981 bp was identified (Figure 1A). The *DcGSK3* encodes a protein containing 326 amino acids and has a predicted molecular weight of 36.71 kDa and a theoretical isoelectric point of 7.08. The structural domain analysis revealed that DcGSK3 consisted of one Pkinase_Tyr domain at the N-terminal and two low complexity (LC) domains at the C-terminal (Figure 1B). The phylogenetic tree analysis showed that the DcGSK3 is more closely related to the glycogen synthase kinase from Hemiptera (*A. pisum*, *M. persicae* and *A. glycines*) than its counterparts from Lepidoptera, Diptera, Coleoptera and Hymenoptera (Figure 1C). A multiple alignment analysis revealed that the DcGSK3 protein sequence shares a high level of identity with *A. pisum* and *M. persicae* (Figure 2A). Further, 13 conserved phosphorylation sites were found when comparing the DcGSK3 with the GSK3 sequence from *A. pisum* and *M. persicae* (Figure 2A). By using the SWISS-MODEL software, the tertiary structure of DcGSK3 was predicted. The results showed that the DcGSK3 protein is primarily composed of alpha helices (Figure 2B).

### 2.2. Expression Patterns of DcGSK3 at Different Tissues and Developmental Stages

The expression patterns of *DcGSK3* in different tissues and at different developmental stages were analyzed by RT-qPCR. The results showed that *DcGSK3* expression was detected in all tissues, including the midgut, leg, head, wing, integument and fat body (Figure 3). However, the highest expression level of *DcGSK3* was found in the head, whereas it was low in the midgut. The expression level of *DcGSK3* in the head was 6.3 times higher than in the midgut. In addition, the expression level of *DcGSK3* was significantly downregulated from the egg to the nymph stages. However, the expression level of *DcGSK3* was constant without significant variations from the first-instar nymph to the adult stage (Figure 3).

### 2.3. Analysis of D. citri Phenotype and DcGSK3 Expression Level after RNAi

In order to confirm the functions of *DcGSK3* in the development of *D. citri*, the *DcGSK3* expression level and effects on the *D. citri* phenotype were investigated following a microinjection of ds*DcGSK3*. The results showed that the *DcGSK3* expression level was significantly downregulated at 24 h and 48 h after the injection of dsRNA (Figure 4B). The transition from a fifth-instar nymph to an adult was disrupted in the ds*DcGSK3* treatment group, and the emerged adult showed two abnormal phenotypes. The first phenotype was that fifth-instar nymphs could not undergo a normal molt. They also had abnormal dorsal tergites and malformed wings. In the second phenotype, the fifth-instar nymphs were capable of molting, but their wings were irregular and curled at the distal end or were small in size. However, the nymphs in the control group (ds*GFP* treatment) were able to molt normally (Figure 4C).

The mortality and malformation rates were significantly upregulated from 12 h to 48 h after the silencing of *DcGSK3*. The mortality rate in the treatment group (ds*DcGSK3*) was 18.6% while it was 4% in the control group (ds*GFP*) at 12 h, and the mortality reached 71.5% at 48 h (Figure 5A). However, the mortality rate was consistently below 20% from 12 h to 48 h in the control group. *D. citri* had no obvious abnormal phenomenon at 12 h after treatment with ds*GFP* or ds*DcGSK3*. Nevertheless, the malformation rate increased rapidly from 8% at 24 h to 32% at 48 h in the ds*DcGSK3* groups (Figure 5B). However, the malformation rate was increased slightly in the ds*GFP* groups, but the difference was not significant. In contrast, the rate of the cumulative molting from 12 h to 48 h was increased after the injection of ds*GFP* and ds*DcGSK3* (Figure 5C). In the ds*DcGSK3* groups, the cumulative molting rate at 48 h reached 34%, while it was approximately 15% in the ds*GFP* groups. These results indicated that the silencing of *DcGSK3* impaired the molting process of fifth-instar *D. citri* nymphs.

### 2.4. Analysis of Trehalase Activity and Trehalose, Glycogen, Glucose and Free Fatty Acid Contents after Silencing of DcGSK3

In order to determine whether the *DcGSK3* gene can regulate sugar metabolism and fatty acid synthesis, the trehalase activity, trehalose, glucose, glycogen and free fatty acid contents were measured at 24 h and 48 h after the silencing of *DcGSK3*. The results showed that the membrane-bound trehalase activity decreased at 24 h, while it had no significant change between the treatment group (ds*DcGSK3* treatment) and control group (ds*GFP* treatment) (Figure 6A). On the contrary, the soluble trehalase activity increased at 24 h; however, it had no obvious change at 48 h after silencing *DcGSK3* (Figure 6B). Following the inhibition of *DcGSK3* expression, glucose, trehalose and glycogen content were reduced at 24 h and 48 h (Figure 6C–E). Furthermore, the free fatty acid content was also downregulated at 24 h in the treatment group (ds*DcGSK3* treatment) compared with the control group (ds*GFP* treatment) (Figure 6F). These results suggest that *DcGSK3* regulates the metabolism of sugar and fatty acids in *D. citri*.

### 2.5. Analysis of Chitin-Metabolism-Related Genes after Silencing of DcGSK3

We performed an RT-qPCR to detect the expression levels of chitin-metabolism-related genes, including *TPS1*, *Tre1-1*, *Tre1-2*, *Tre2*, *HK*, *G6PI*, *GFAT*, *GNPNA*, *PAGM*, *UAP*, *CHT* and *NAG*, to determine whether *DcGSK3* could regulate *D. citri* chitin metabolism. The results showed that the relative expression levels of *D. citri Tre1-1*, *Tre1-2*, *Tre2*, *HK* and *G6PI* were downregulated at 24 h and 48 h after silencing *DcGSK3* (Figure 7). Further, the expression levels of the three genes decreased only at 48 h after treatment with *DcGSK3*, whereas no significant difference was detected between the ds*DcGSK3* groups and ds*GFP* groups at 24 h, including *TPS1*, *GNPNA* and *PAGM* (Figure 7). The relative expression levels of *GFAT* and *NAG* were upregulated in the ds*DcGSK3* compared with the ds*GFP* group at 24 h, with no significant difference at 48 h (Figure 7). Additionally, the expression level of *UAP* was downregulated at 24 h and exhibited upregulation at 48 h after the silencing of *DcGSK3*, and the *CHT* expression level had no significant difference at 24 h and 48 h between the ds*DcGSK3* groups and ds*GFP* groups (Figure 7). These results indicated that *DcGSK3* might play an important role in regulating chitin metabolism in *D. citri*.

### 2.6. Analysis of Fatty Acid Metabolism-Related Genes after Silencing of DcGSK3

The acetyl-CoA produced from glycolysis can be utilized to form lipids. In order to analyze the effect of *DcGSK3* on *D. citri* fatty acid metabolism, a total of six genes involved in the synthesis and degradation of fatty acids were analyzed at 24 h and 48 h after the silencing of *DcGSK3*. The results suggested that two genes associated with fatty acid synthesis were significantly downregulated at 24 h after the knockdown of *DcGSK3*. In contrast, there was no significant difference between the ds*DcGSK3* group and ds*GFP* group at 48 h, including *acetyl-CoA carboxylase-like* (*ACC*) and *fatty acid synthase-like* (*FAS*) products (Figure 8). In addition, three genes involved in the oxidative degradation of fatty acids exhibited similar expression patterns. They were upregulated at 24 h or 48 h after silencing *DcGSK3*, including *medium-chain-specific acyl-CoA dehydrogenase* (*MCAD*), *glutaryl-CoA dehydrogenase* (*GCD*) and *Lipase*. The expression level of *acetyl-CoA acetyltransferase* (*ACAT*) remained unchanged at 24 h and 48 h after silencing *DcGSK3* (Figure 8). We speculated that *DcGSK3* might be involved in regulating fatty acid metabolism in *D. citri*.

### 2.7. Transcriptome Sequencing and Reads Assembly

After removing the redundant reads, a total of 40,406,422 (92.4%), 43,941,144 (96.2%) and 42,166,692 (93.4%) clean reads from the ds*DcGSK3* groups at 24 h were obtained; 41,587,526 (94.1%), 40,357,952 (93.5%) and 41,782,344 (93.9%) clean reads from the ds*GFP* groups at 24 h were obtained; 41,535,154 (91.1%), 38,889,614 (94.9%) and 42,260,254 (92.0%) clean reads from the ds*DcGSK3* groups at 48 h were obtained; and 40,019,100 (91.6%), 43,957,846 (91.9%) and 40,646,136 (99.9%) clean reads from the ds*GFP* groups at 48 h were obtained. The values of Q20 and Q30 were approximately 97% and 92%, respectively. The values of the GC content in different samples were around 40% (Appendix A). Furthermore, 31,784,571 (78.47%), 34,344,703 (78.16%) and 32,592,895 (77.3%) clean reads from the ds*DcGSK3* groups at 24 h; 32,269,610 (77.59%), 31,432,526 (77.88%) and 32,418,487 (77.59%) clean reads from the ds*GFP* groups at 24 h; 32,145,689 (77.39%), 30,270,911 (77.84%) and 33,192,001 (78.54%) clean reads from the ds*DcGSK3* groups at 48 h; and 31,671,159 (79.14%), 33,878,922 (77.07%) and 31,293,510 (76.99%) clean reads from the ds*GFP* groups at 48 h were successfully mapped to the *D. citri* genome (Appendix A).

### 2.8. Identification of DEGs and Enrichment Analysis

The DEGs were identified in comparable groups by using the DESeq method. A total of 991 DEGs were identified in the ds*DcGSK3* groups compared with the ds*GFP* groups at 24 h, of which 444 DEGs were upregulated and 547 DEGs were downregulated (Figure 9A; Appendix A). A total of 1692 DEGs were identified in the ds*DcGSK3* groups compared with the ds*GFP* groups at 48 h, among which 1020 DEGs were upregulated and 672 DEGs were downregulated (Figure 9B; Appendix A).

A GO enrichment analysis suggested that most DEGs were mainly involved in the carbohydrate metabolic process at 24 h after the knockdown of *DcGSK3* (Figure 10A). The upregulated DEGs were associated with a cellular amide metabolic process and amide biosynthetic process, and the downregulated DEGs were related to transporter activity and the carbohydrate metabolic process. At 48 h between the ds*DcGSK3* treatment groups and ds*GFP* treatment groups, most DEGs were mainly associated with transition metal ion binding and transporter activity. The upregulated DEGs were related to transporter activity and iron ion binding, while the downregulated DEGs were mainly associated with the ribosome’s structural molecular activity and structural constituent (Figure 10B). A KEGG enrichment revealed that most DEGs were significantly enriched for starch, sucrose metabolism and fatty acid metabolism at 24 h after the knockdown of *DcGSK3* (Figure 11A). At 48 h between the ds*DcGSK3* treatment groups and ds*GFP* treatment groups, most DEGs were significantly enriched for lysosome, ribosome and fatty acid metabolism (Figure 11B).

### 2.9. Identification of DEGs Involved in Chitin and Fatty Acid Metabolism

Based on the transcriptome analysis, numerous DEGs involved in chitin and fatty acid metabolism were altered in different comparable groups. In the chitin metabolism, 16 DEGs related to chitin metabolism were identified based on transcriptome data, of which 13 (81.2%) DEGs were downregulated at 24 h in the ds*DcGSK3* groups compared with the ds*GFP* groups (Figure 12; Table 1). Furthermore, three genes were upregulated at 24 h after the silencing of *DcGSK3*, including *IDGF2*, *PAGM* and *HK2*. It was found that 11 DEGs were upregulated and 5 DEGs showed downregulation at 48 h after the silencing of *DcGSK3*. These results indicated that the inhibition of *DcGSK3* at 24 h significantly disrupted *D. citri* chitin metabolism and that the effect could be recovered at 48 h.

For fatty acid metabolism, a total of 16 DEGs involved in fatty acid metabolism were identified, among which 11 DEGs were downregulated at 24 h in the ds*DcGSK3* groups compared with the ds*GFP* groups (Figure 13A; Table 2). Moreover, four DEGs were upregulated at 24 h, including *Lipase 3*, *FA2H*, *ELOVL1* and *Lipase 1* (Figure 13A). At 48 h after silencing *DcGSK3*, nine DEGs were upregulated and four DEGs were downregulated. The relative expression levels of 16 DEGs associated with fatty acid were further validated by RT-qPCR (Figure 13B).

## 3. Discussion

This study identified a *GSK3* gene with the complete ORF sequence from the *D. citri* genome database. A bioinformatics analysis revealed that DcGSK3 contains a Pkinase_Tyr domain and two LC domains. Moreover, DcGSK3 comprises 13 phosphorylation sites, liking serine, threonine and tyrosine modification sites. The GSK3 enzyme plays a pivotal role in the β-catenin signaling pathway, which is responsible for the phosphorylation and degradation of β-catenin through ubiquitin proteasome [33]. The results indicated that *DcGSK3* is a kinase that is involved in regulating the downstream signaling pathways. In *Drosophila*, a homolog of GSK3 named Shaggy regulates development by controlling Wnt/β-catenin signaling [34]. In *Tribolium castaneum*, the Wnt/β-catenin signaling pathway integrates the patterning and metabolism of the insect growth zone [35]. Chen et al. revealed that Wnt/β-catenin signaling regulates pupal development by upregulating c-Myc and AP-4 in *Helicoverpa armigera* [36]. Consequently, we speculated that *DcGSK3* might regulate Wnt/β-catenin signaling to influence the growth and development of *D. citri*.

The relative expression level of *DcGSK3* was analyzed in different tissues and developmental stages. The results indicated that *DcGSK3* expression was highest in the head of *D. citri*, followed by the wing. In insects, the head tissue is central to the brain and is responsible for hormone secretion and chemosensory response. GSK3 is involved in the regulation of insulin. The binding of insulin to the corresponding receptor can activate the tyrosine kinase activity of the receptor, and further regulate phosphoinositol-3 kinase (PI3K), 3-phosphoinositide-dependent protein kinase (PDK1) and protein kinase B (PKB) to induce the phosphorylation of the GSK3 substrates [37]. In *Drosophila*, a total of eight insulin-like peptides (ILPs) have been identified, among which ILPs 2, 3 and 5 are produced by neurosecretory cells (NCSs) in the brain [38]. We also found that *DcGSK3* exhibited a high expression in the wing. Intriguingly, Xu et al. reported that two insulin receptors regulate wing morphs in *N. lugens* [37]. Accordingly, we considered that the insulin-related peptides secreted from the *D. citri* brain regulate the downstream crosstalk between the insulin signal pathways and sugar metabolism pathways modulated by *DcGSK3*. *DcGSK3* was expressed at a higher level in the egg at different development stages. By acting as a downstream regulatory switch, GSK3 controls the output of numerous signaling pathways induced by various stimuli [39]. In *Drosophila*, a membrane-tethered form of GSKS activates Wnt signaling, thereby regulating embryonic development [40]. The expression of *GSK3* was found to be restricted to the embryonic tissue of *T. castaneum*, which indicates that embryonic and extra-embryonic cells display different metabolic activities [41]. In the hemipteran *Rhodnius prolixus*, the knockdown of *GSK3* induced an impaired oogenesis and early embryogenesis, suggesting that GSK3 is essential for both processes in *R. prolixus* [42]. Hence, we speculated that *DcGSK3* might play an important role in *D. citri* embryonic development.

Trehalose is the major blood sugar in insects and plays a crucial role in providing an instant energy source and responding to abiotic stresses [43]. Trehalose is synthesized by the catalysis of trehalose-6-phosphate synthase (TPS) and is degraded by the catalysis of trehalase [44]. The synthetic substrates of trehalose, uridine diphosphate glucose and phosphate-6-glucose are derived from the decomposition of glycogen [45]. Therefore, trehalose metabolism and glycogen utilization are closely related. The present study aimed to determine whether *DcGSK3* plays a role in the growth and development of *D. citri* through an RNAi bioassay. Even though RNAi is one of the most powerful tools for functional gene research and the control of agricultural pests, there are a few potential challenges associated with the lack of effective dsRNA delivery methods [23]. Particularly for *D. citri*, dsRNA delivery mainly relied on topical feeding, soaking and artificial feeding [46,47,48]. However, dsRNA delivery by microinjection has not been reported in *D. citri*. In this study, *DcGSK3* was effectively silenced at 24 h and 48 h after the injection of dsRNA. By silencing *DcGSK3*, the *D. citri* molting from a fifth-instar nymph to an adult was significantly disrupted. According to Ding et al., the knockdown of *GSK3* significantly affected the molting and wing formation of *N. lugens* [20].

Chitin is a major structural component of the insect exoskeleton and plays an important role in molting [49]. Therefore, we considered that the abnormal molt observed after the inhibition of *DcGSK3* may have resulted from a chitin metabolism block in *D. citri*. The pathway of insect chitin biosynthesis begins with trehalose and is catalyzed by hexokinase (HK), glucose-6-phosphate isomerase (G6PI), fructose 6-phosphate transaminase (GFAT), glucosamine-6-phosphate *N*-acetyltranferase (GNPNA), phosphor acetylglucosamine mutase (PAGM), UDP-*N*-acetylglucosamine pyrophosphorylase (UAP) and chitin synthase (CHS) [8]. We found that the silencing of *DcGSK3* significantly reduced the expression levels of chitin-metabolism-related genes at 24 h and 48 h. Furthermore, membrane-bound trehalase activity was significantly reduced at 24 h after the knockdown of *DcGSK3*, while soluble trehalase activity was increased significantly. In our previous research, three trehalase genes were identified based on the *D. citri* genome database, including *Tre1-1*, *Tre1-2* and *Tre2*, of which the Tre1s were related to soluble trehalase and Tre2 was associated with membrane-bound trehalase [31]. Here, we also found that the expression levels of *Tre1-1*, *Tre1-2* and *Tre2* were downregulated at 24 h and 48 h after silencing *DcGSK3*. These results suggest that *DcGSK3* may regulate chitin metabolism by influencing *D. citri* Tres expression levels. Intriguingly, a transcriptome analysis revealed that the genes related to chitin metabolism were downregulated at 24 after the knockdown of *DcGSK3*, whereas they were upregulated at 48 h. The fifth-instar nymph of *D. citri* is an important developmental stage in the transition from nymph to adult [48]. In the early stage of *D. citri* molting, chitin synthesis and degradation are essential for forming a new cuticle. *D. citri* molt was significantly influenced by genes that inhibit chitin metabolism. Nonetheless, *D. citri* needs to increase the expression level of the genes related to chitin metabolism in the late stage of molt to produce more chitin by a regulatory feedback mechanism. Additionally, two transport genes were identified as being involved in glucose and trehalose transport, including *facilitated glucose transporter membrane 6* (*GLUT6*) and *facilitated trehalose transporter 1* (*TRET1*). Sugar transporters play an important role in controlling carbohydrate transport in organisms and are responsible for mediating the movement of sugars into cells [50]. The results showed that *GLUT6* and *TRET1* expression levels significantly decreased at 24 h following *DcGSK3* silencing, indicating that *DcGSK3* might indirectly regulate *GLUT6* and *TRET1* expression levels to influence the transport of glucose and trehalose into cells.

Free fatty acids (FFAs) are a diverse group of lipids that play an important role in maintaining homeostasis and managing cellular processes. In insects, FFAs are essential components of the cell and cell membrane, as well as sources of energy [51]. The energy reserves of insects are stored in the form of glycogen and triglycerides. Fatty acids stored as triglyceride can also be used for energy production through β-oxidation [14]. This study found that free fatty acid content was downregulated at 48 h after silencing *DcGSK3*. Furthermore, the expression levels of fatty-acid-synthesis-related genes were downregulated, while the genes associated with fatty acid degradation were upregulated after silencing *DcGSK3*. We speculated that the inhibition of *DcGSK3* reduced energy production and promoted fatty acid degradation to compensate energy shortages. Based on transcriptome sequencing results, 16 DEGs involved in fatty acid metabolism were identified, of which 12 genes were downregulated at 24 h in the treatment groups (ds*DcGSK3* treatment) compared with the control groups (ds*GFP* treatment). In addition, four genes were upregulated at 24 h after silencing *DcGSK3*, including *lipase 3*, *FA2H*, *ELOVL1* and *lipase 1*. Lipases are ubiquitous enzymes in nature that play a crucial role in fat metabolism by catalyzing the hydrolysis of triacylglycerol to free fatty acids and glycerol [52]. The results showed that the expression levels of lipase1 and lipase 3 were significantly reduced after the knockdown of *DcGSK3*. We speculated that the inhibition of *DcGSK3* significantly reduced the glucose and trehalose contents in *D. citri* and then lipase-related genes were activated to promote the hydrolysis of triacylglycerol. Further research is needed to determine the specific molecular mechanisms involved.

## 4. Materials and Methods

### 4.1. D. citri Rearing and Sample Collection

The *D. citri* adults were collected from Ganzhou China and were continuously reared on *Murraya exotica* inside steel cages (60 × 60 × 90 cm^3^) at Gannan Normal University, China. *D. citri* rearing was conducted at a constant temperature (27 ± 1 °C), relative humidity (75 ± 5%) and with a 16:8 (light: dark) photoperiod. To maintain the consistent growth and development of *D. citri*, the females, after mating, were placed into the flourishing *M. exotica*. After 24 h, all the *D. citri* adults were removed using a portable aspirator and the eggs continued to hatch and develop into nymphs. Using a stereomicroscope, different stages of *D. citri* were obtained based on morphological characteristics. All the collected samples were kept at −80 °C. Each group of samples contained three biological replicates.

### 4.2. Identification of DcGSK3 and in Silico Analysis

The protein sequences of GSK3 from *N. lugens* (XP_039293499), *Bombyx mori* (XM_004928285) and *Drosophila melanogaster* (NP_476714) were downloaded from the National Center for Biotechnology Information (NCBI) (https://www.ncbi.nlm.nih.gov/, accessed on 12 January 2022) and these GSK3 protein sequences were used as queries for blasting against the *D. citri* genome database (https://citrusgreening.org/ftp/genomes/Diaphorina_citri/assembly/DIACI_v2.0/, accessed on 12 January 2022). The candidate GSK3 sequences were analyzed to obtain the complete *DcGSK3* cDNA sequence. A complete open reading frame of *DcGSK3* was amplified by PCR. The purified product was linked to the pMD19-T vector and sequenced by a biotechnology company (Sangon Biotech, Shanghai, China).

The amino acid sequence of *DcGSK3* was analyzed using DNASTAR software. The theoretical molecular weights (MWs) and isoelectric points (pIs) of DcGSK3 were calculated with ExPASy online software (http://web.expasy.org/compute_pi, accessed on 1 May 2021). The conserved domain was identified by using SMART online software (http://smart.embl-heidelberg.de/, accessed on 20 May 2021) and Illustrate for Biological Sequences (IBS) software (http://ibs.biocuckoo.org/, accessed on 1 May 2021). The multiple sequence alignment was performed using DNAMAN software. The SWISS-MODEL software (https://swissmodel.expasy.org/interactive, accessed on 1 January 2022) was used to predict the sense structure of the protein. MEGA 7.0 software was used for constructing the phylogenetic tree with the neighbor-joining method with 1000 replicates [53].

### 4.3. RNA Isolation, cDNA Synthesis and RT-qPCR Analysis

The total RNA was isolated from *D. citri* in different tissues (midgut, leg, head, wing, integument and fat body) and at different developmental stages (egg, 1st-, 2nd-, 3rd-, 4th- and 5th-instar nymphs and adults) using the animal tissue total RNA kit (Simgen). RNA quality and quantity were measured with a NanoDrop 2000 spectrophotometer (Thermo Fisher Scientific, New York, NY, USA). The total RNA from different samples was reverse transcribed in a 20 μL reaction system using a cDNA synthesis master mix kit according to a previous protocol [54]. All the cDNA samples were diluted to the same concentration as a template for RT-qPCR analysis. RT-qPCR reactions were conducted using a LightCycler^®^96 PCR Detection System (Roche, Basel, Switzerland). The specific sample addition procedures were performed according to a previous report [44]. The relative expression levels were analyzed using the 2^−∆∆Ct^ method. The constitutively expressed *glyceraldehyde-3-phosphate dehydrogenase* (*GAPDH*) gene was used as a reference gene. All primers were presented in Appendix A. There were three biological and technical replicates for each sample.

### 4.4. Detection of Trehalase Activity, Sugar and Free Fatty Acid Content

The trehalase activity was assayed based on a previous protocol [31]. After treatment with ds*DcGSK3* and ds*GFP*, a total of 30 5th-instar nymphs were added to a 1.5 mL centrifuge tube and then 1 mL precooled PBS (pH 7.0) was added. The mixture was homogenized on ice using a glass homogenizer and the homogenate was centrifuged at 1000× *g* for 20 min at 4 °C. Approximately 400 μL of supernatant was collected for trehalase activity analysis. The remaining supernatant was used to determine the trehalase, glucose and glycogen contents. The 400 μL of supernatant was centrifuged at 20,800× *g* for 60 min at 4 °C. The ultracentrifugal supernatant was obtained to determine the soluble trehalase activity and the pellet was suspended with PBS (pH 7.0) to measure membrane-bound trehalase activity. Finally, the soluble and membrane-bound trehalase activities were assayed using a glucose assay kit (Simgen, Hangzhou, China).

For trehalose measurement, the suspension was incubated with trehalase (Sigma-Aldrich) at 37 °C overnight. The free glucose calculated from the sample without an amyloglucosidase or trehalase treatment was subtracted from the glycogen or trehalose value, respectively. For the glucose measurement, 30 nymphs were homogenized in 300 μL of PBS before measuring the glucose levels with the GO reagent (Sigma-Aldrich, St. Louis, MO, USA). The protein concentration was determined using a BCA Protein Assay Reagent (Pierce Chemical Company, Rockford, IL, USA) according to the manufacturer’s instructions. In order to measure glycogen, the suspension was incubated for 1 h with amyloglucosidase (catalog no. 10115, Sigma-Aldrich) at 50 °C.

Free fatty acids were determined from the ds*GFP-* and ds*DcGSK3*-treated samples according to the protocol reported by Yang et al. with minor modifications [55]. Briefly, 40 *D. citri* were added to a 1.5 mL centrifugation tube and 1 mL of extracting solution was added before homogenizing on ice. All the samples were centrifuged at 8000× *g* for 10 min at 4 °C and the supernatants were collected. The protein concentrations from different samples were determined using a BCA Protein Assay Reagent (Pierce Chemical Company, Rockford, IL). A mother solution containing 5 μmol/mL of palmitic acid was diluted to different concentrations (0.05, 0.1, 0.2, 0.4, 0.6, 0.8 and 1.0 μmol/mL) as a standard substance. A standard curve was constructed using an FFA content determination kit (Solarbio, Beijing, China) with the above standard substance at an absorbance of 550 nm. Finally, the contents of FFA from different samples were analyzed based on three biological replicates.

### 4.5. dsRNA Synthesis and Microinjection

The RNAi-based silencing of *DcGSK3* gene expression was carried out to analyze the functions of *DcGSK3* on *D. citri*. The ds*DcGSK3* and ds*GFP* were synthesized using the T7 RioMAXTM Express RNAi System (Promega, Madison, WI, USA) according to the manufacturer’s instructions. The specific ds*DcGSK3* and ds*GFP* primers containing a T7 promoter sequence were designed and are listed in Appendix A. The synthetic ds*DcGSK3* was diluted to 600 ng/mL of working solution using RNase-free water and a moderate 0.1% red food dye was added as an indicator. For the RNAi experiment, 300 fifth-instar *D. citri* nymphs were divided into three groups. A sticker was attached on the dorsum to maintain the ventrum in an upright position for injection (Figure 4A). A total of 20 nL of ds*DcGSK3* was injected into the fifth-instar *D. citri* nymphs and then transferred onto the fresh *M. exotica* seedlings. An equal amount of ds*GFP* was injected as a control. All the experiments contained three biological replicates. All the alive *D. citri* were collected at 24 h and 48 h after dsRNA treatment. The efficiency of silencing was determined using an RT-qPCR as described above.

### 4.6. cDNA Library Preparation and Illumina Sequencing

Transcriptome sequencing was carried out at Novogene Biological Information Technology Co., Ltd. (Tianjin, China). Approximately 100 *D. citri* were collected from each treatment group (treated with ds*DcGSK3*) and control group (treated with ds*GFP*) at 24 h and 48 h following the injection of dsRNA. All samples contained three biological replicates. The concentration and purity of the RNA were measured by a Qubit RNA Assay Kit (Life Technologies, CA, USA). For cDNA library construction, a total of 1 μg of RNA was used for constructing a cDNA library with the TruSeq RNA Sample Preparation Kit v2 (Illumina, San Diego, CA, USA). The prepared cDNA library was sequenced by the Illumina HiSeq platform, generating 150 bp paired-end reads. The clean reads were obtained by removing reads that contained the adapter from raw data. Furthermore, the Q20, the Q30 and the GC content of the clean data were calculated.

### 4.7. Transcriptome Analysis after Silencing of DcGSK3

The transcriptome data were mapped to a *D. citri* reference genome (https://www.citrusgreening.org/, accessed on 12 January 2022) using a Hisat2 (version 2.0.5; https://anaconda.org/biobuilds/hisat2, accessed on 12 January 2022) aligner. This generated a database of splice junctions based on the gene model annotation file. The expression levels of the genes were calculated using reads per kilobase per million mapped reads (RPKM). Differential expression analyses of the genes between the ds*GFP* and ds*DcGSK3* groups at 24 h and 48 h were performed using the DESeq2 R package. The *p*-values were adjusted using the Benjamini–Hochberg method to control for the false-discovery rate. A corrected *p*-value of 0.05 and an absolute |log2 (fold change)| of 0 was set as the thresholds for significantly different gene expression. The hierarchical cluster analysis of DEGs was conducted using Genesis software. The topGO R package, which implements the Gene ontology (GO) terms, was used for the enrichment analysis of length-corrected DEGs. A Kyoto Encyclopedia of Genes and Genomes (KEGG) pathway enrichment analysis for the DEGs was performed using KOBAS. A *p*-value of <0.01 was set as the threshold.

### 4.8. Statistical Analysis

In this study, all data were analyzed using a one-way analysis of variance and Turkey’s test. *p*-values of less than 0.05 and 0.01 were defined as significant and extremely significant, respectively.

## 5. Conclusions

In summary, we identified a *GSK3* gene with the complete ORF from the *D. citri* genome database and named it *DcGSK3*. *DcGSK3* showed a high expression in the head and egg. Furthermore, the inhibition of *DcGSK3* by RNAi led to an abnormal phenotype and downregulated the synthesis of glucose, trehalose, glycogen and free fatty acid contents. Moreover, an RT-qPCR and comparative transcriptome sequencing analysis revealed that the silencing of *DcGSK3* significantly suppressed the expression levels of the genes involved in chitin and fatty acid metabolism. In addition to providing a foundation for further research on *DcGSK3*, these results also provide a novel target for controlling *D. citri*.

## Figures and Tables

**Figure 1 ijms-23-09654-f001:**
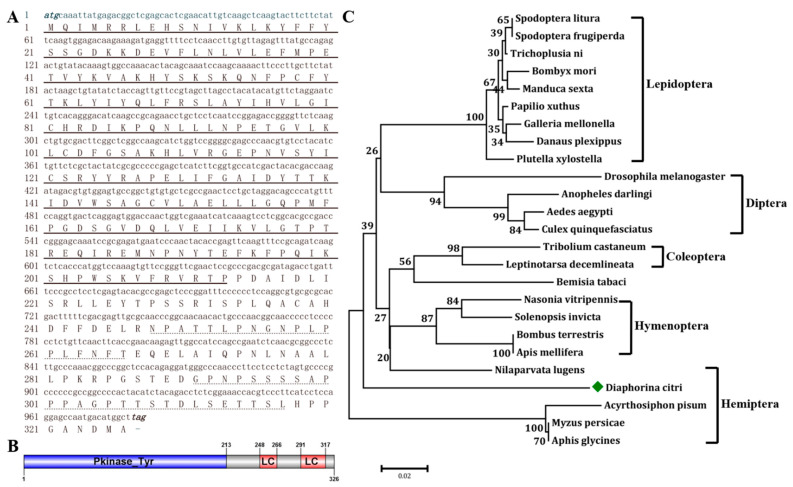
Bioinformatics analysis of DcGSK3. (**A**) The nucleotide and amino acid sequence of *DcGSK3* gene. Numbers on the left side represent nucleotide and amino acid positions. The black solid line indicates Pkinase_Tyr domain. The black dotted line indicates low complexity domain. (**B**) The structural domain analysis of DcGSK3 protein. The blue region indicates the Pkinase_Tyr domain and the red region indicates the low complexity (LC) domain. (**C**) Phylogenetic tree analysis of *D. citri* GSK3 with other insect species. The green rhombus indicates the DcGSK3.

**Figure 2 ijms-23-09654-f002:**
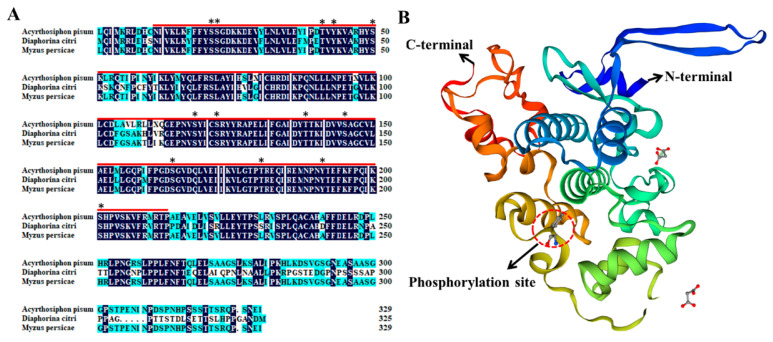
Multiple sequence alignment and protein structure analysis of DcGSK3. (**A**) Multiple sequence alignment of the GSK3 from three insect species, including *Diaphorina citri*, *Acyrthosiphon pisum* and *Myzus persicae*. The conserved amino acid residues are highlighted in blue, and similar amino acid residues were labeled in turquoise. The asterisk indicates the phosphorylation site. (**B**) Protein structure analysis of DcGSK3 by SWISS-MODEL software (Version 3.5, https://swissmodel.expasy.org/).

**Figure 3 ijms-23-09654-f003:**
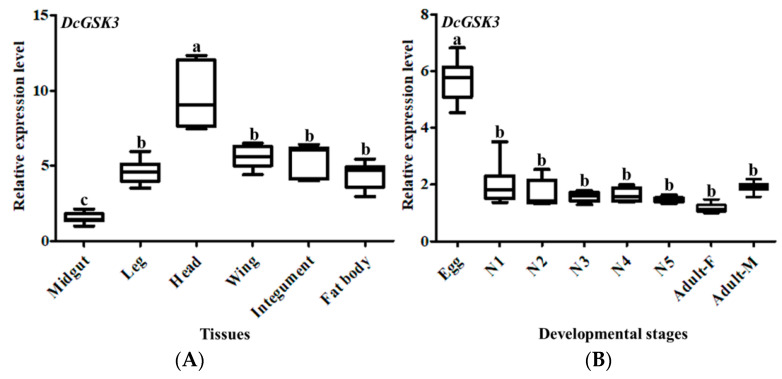
The relative expression levels analysis of *DcGSK3* in different tissues (**A**) and different developmental stages (**B**). Data were normalized using *glyceraldehyde-3-phosphate dehydrogenase* (*GAPDH*) and are represented as the means ± standard errors of the means from three independent experiments. Different lowercase letters indicate the significant differences, for example, a, b and c (*p* < 0.05).

**Figure 4 ijms-23-09654-f004:**
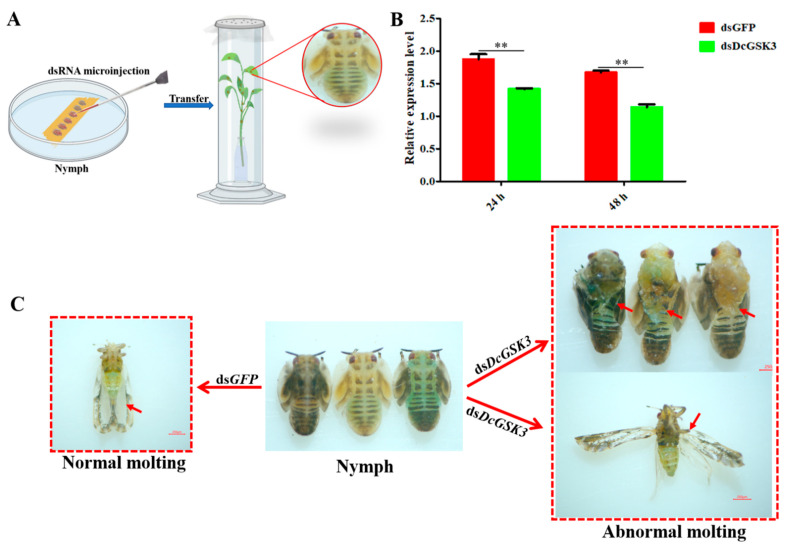
Detection of *DcGSK3* expression levels and phenotypic observation after treatment with ds*DcGSK3* and ds*GFP*. (**A**) Illustration of the protocol used for RNA interference. (**B**) Analysis of *DcGSK3* expression levels after treatment with ds*DcGSK3* and ds*GFP*. The asterisks indicate the significance differences by ** *p* < 0.01. (**C**) Phenotypic observation of *D. citri* adult at 48 h.

**Figure 5 ijms-23-09654-f005:**
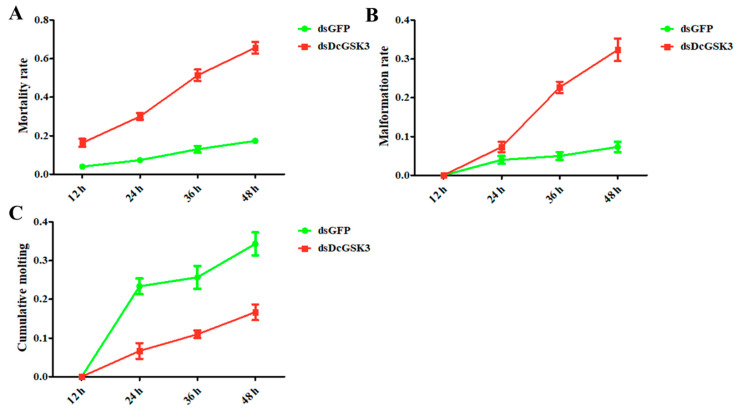
Effect of *DcGSK3* on the development of *D. citri* from 12 h to 48 h. (**A**) Detection of mortality after inhibition of *DcGSK3*. The ds*GFP* treatment group was used as a control; (**B**) detection of malformation rate after inhibition of *DcGSK3*; (**C**) detection of cumulative molting of *D. citri* after inhibition of *DcGSK3*.

**Figure 6 ijms-23-09654-f006:**
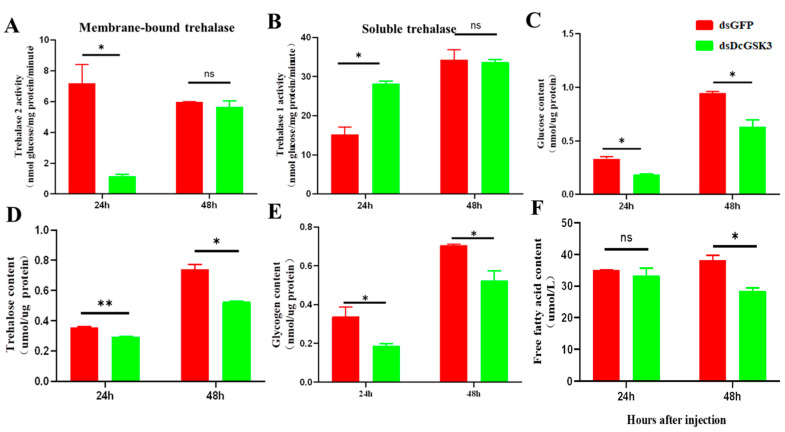
Effect of silencing *DcGSK3* on the trehalase activity, glucose content, trehalose content, glycogen content and free fatty acid content at 24 h and 48 h. Membrane-bound trehalase activity (**A**), soluble trehalase activity (**B**), glucose content (**C**), trehalose content, (**D**) glycogen content (**E**) and free fatty acid content (**F**) at 24 h and 48 h. Each experiment contained three biological replicates. The significant differences are indicated by * (*p* < 0.05) or ** (*p* < 0.01). The “ns” indicated no significant differences.

**Figure 7 ijms-23-09654-f007:**
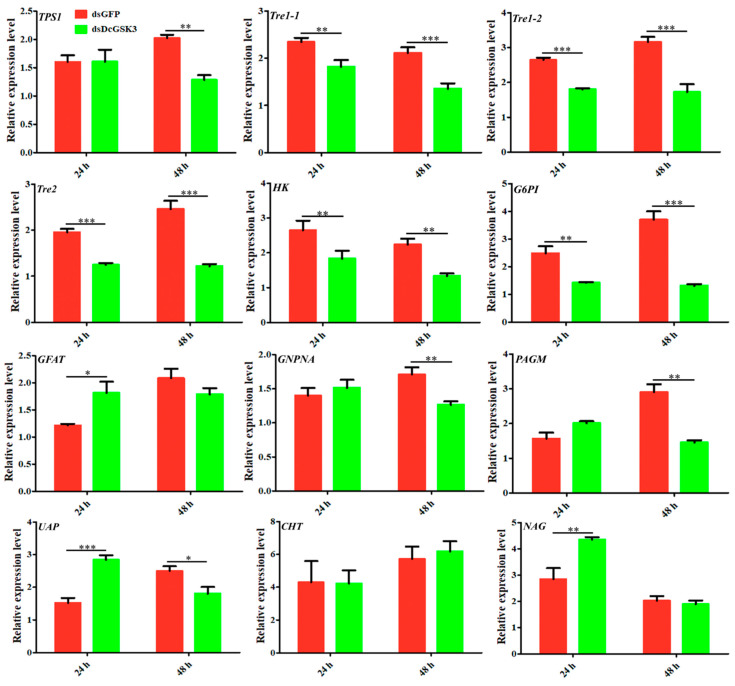
Effects of silencing *DcGSK3* on expression levels of twelve genes involved in chitin metabolism in *D. citri*. The ds*GFP* treatment group was used as a control. The mean expression level represents three biological replicates. The asterisks indicate significant differences by * (*p* < 0.05), ** (*p* < 0.01) and *** (*p* < 0.001).

**Figure 8 ijms-23-09654-f008:**
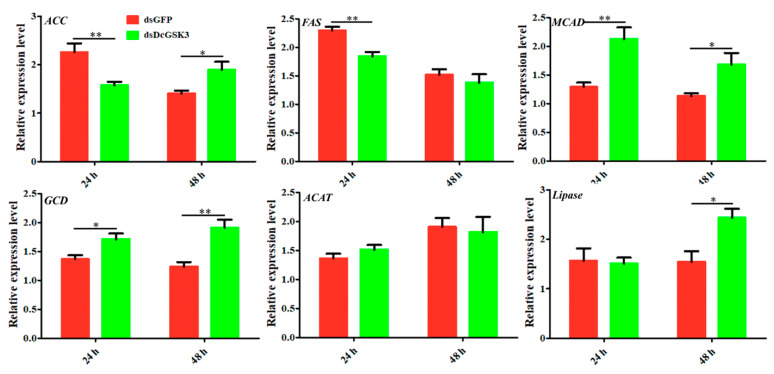
Effects of silencing *DcGSK3* on expression levels of six genes related to fatty acid metabolism in *D. citri*. The ds*GFP* treatment group was used as a control. The mean expression level represents three biological replicates. The asterisks indicate significance differences by * (*p* < 0.05) and ** (*p* < 0.01).

**Figure 9 ijms-23-09654-f009:**
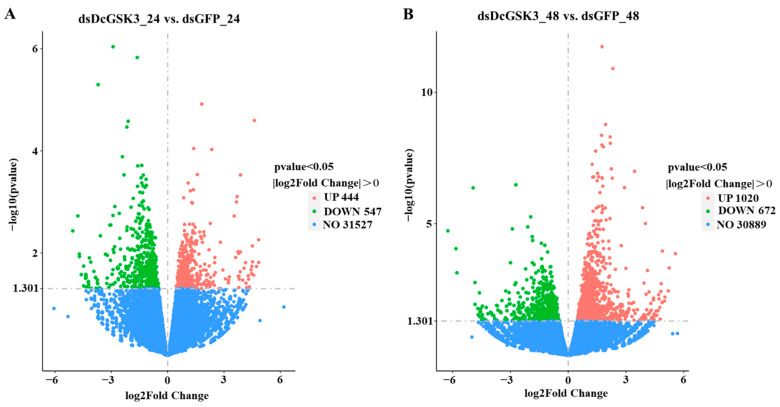
Identification of DEGs at 24 h and 48 h between ds*DcGSK3* groups and ds*GFP* groups. (**A**) Identification of DEGs at 24 h between ds*DcGSK3* groups and ds*GFP* groups. (**B**) Identification of DEGs at 48 h between ds*DcGSK3* groups and ds*GFP* groups. The red and green points represent upregulated genes and downregulated genes, respectively. The blue point indicates no significant difference.

**Figure 10 ijms-23-09654-f010:**
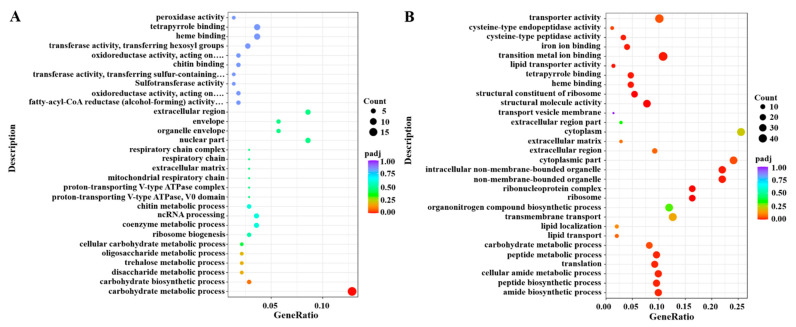
GO enrichment analysis of DEGs between ds*DcGSK3* groups and ds*GFP* groups. (**A**) GO enrichment analysis of DEGs at 24 h. (**B**) GO enrichment analysis of DEGs at 48 h. The size of the bubble indicates the number of DEGs enriched to the corresponding term. The color of the bubble indicates the Q value.

**Figure 11 ijms-23-09654-f011:**
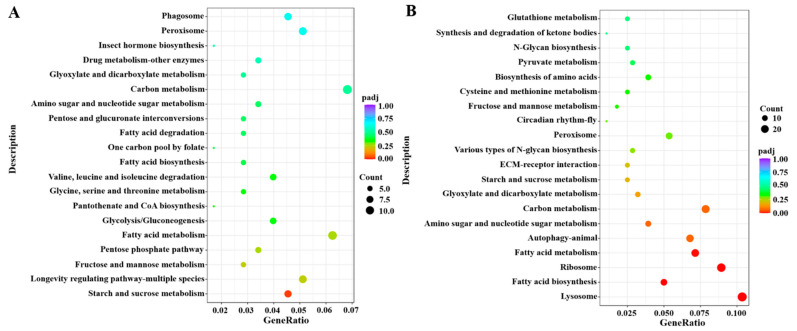
KEGG enrichment analysis of DEGs between ds*DcGSK3* groups and ds*GFP* groups. (**A**) KEGG enrichment analysis of DEGs at 24 h after silencing of *DcGSK3*. (**B**) KEGG enrichment analysis of DEGs at 48 h after silencing of *DcGSK3*. The sizes of the bubble indicate the number of DEGs enriched to the corresponding term. The color of the bubble indicates the Q value.

**Figure 12 ijms-23-09654-f012:**
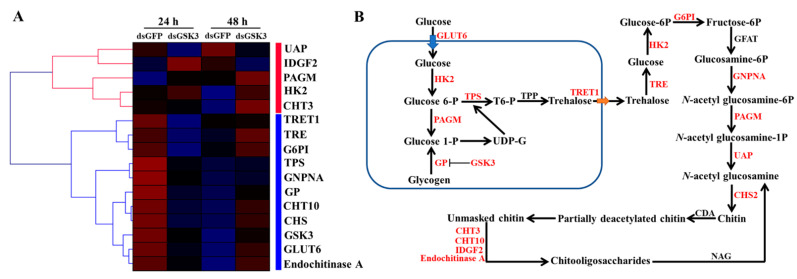
Hierarchical analysis for genes involved in chitin metabolism in different comparable groups. (**A**) Hierarchical analysis of genes related to chitin metabolism at 24 h and 48 h in ds*DcGSK3* groups and ds*GFP* groups. DEG expression is shown with a pseudocolor scale (from −3 to 3), with a red color indicating high expression levels and a bluish-violet color indicating low expression. (**B**) Analysis of chitin metabolism-related genes identified from *D. citri* transcriptome data. The red text indicates that these genes were identified from *D. citri* transcriptome data.

**Figure 13 ijms-23-09654-f013:**
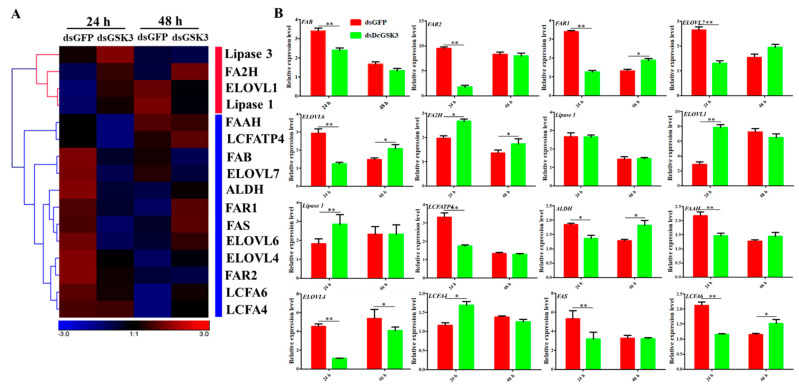
Hierarchical analysis for genes involved in fatty acid metabolism in different comparable groups. (**A**) Hierarchical analysis for DEGs involved in fatty acid metabolism in different comparable groups. DEG expression is shown with a pseudocolor scale (from −3 to 3), with a red color indicating high expression levels and a bluish-violet color indicating low expression. (**B**) The relative expression level analysis of 16 DEGs involved in fatty acid metabolism at 24 h and 48 h after inhibition of *DcGSK3*. The asterisks indicate significance differences by * (*p* < 0.05) and ** (*p* < 0.01).

**Table 1 ijms-23-09654-t001:** Identification of DEGs associated with chitin metabolism in different comparable groups.

Gene ID	Gene Description	24 h	48 h
		Log2FC	Log2FC
103520276	Alpha-trehalose-phosphate synthase	−1.0464	0.1361
103518255	Facilitated trehalose transporter 1	−0.9315	0.0380
103512907	Glycogen phosphorylase	−1.1505	0.6403
103513948	Trehalase	−0.7217	0.5303
103518754	Facilitated glucose transporter member 6	−0.7737	0.9812
103509493	Hexokinase type 2	0.1033	0.4766
103514039	Glucose-6-phosphate isomerase	−0.3089	0.1405
108252063	Glucosamine 6-phosphate *N*-acetyltransferase	−0.3873	0.0833
103507538	Phosphoacetylglucosamine mutase-like	0.1666	0.1141
103524304	UDP-*N*-acetylhexosamine pyrophosphorylase-like protein 1	−0.2398	−0.2047
103505841	Glycogen synthase kinase-3 beta-like	−0.4289	0.5348
103522082	Probable chitinase 10	−0.8906	0.7549
103517759	Chitin synthase chs-2-like	−0.7851	0.6132
108252441	Chitinase-like protein Idgf2	0.2170	−0.1559
103510725	Chitinase-3-like protein	−0.0557	0.3329
113468504	Endochitinase A-like	−0.2582	0.4661

**Table 2 ijms-23-09654-t002:** Identification of DEGs associated with fatty acid metabolism in different comparable groups.

Gene ID	Gene Description	24 h	48 h
		Log2FC	Log2FC
113466381	fatty acid synthase-like	−2.0895	1.5628
103519868	fatty aldehyde dehydrogenase	−0.6424	0.3391
103518946	elongation of very long chain fatty acids protein 4	−0.5674	0.3893
103512002	fatty-acid amide hydrolase 2	−0.5572	−0.1078
103523162	fatty acid-binding protein	−0.5462	−0.3946
113473555	fatty acyl-CoA reductase 2	−0.5322	−0.0643
103514699	fatty acyl-CoA reductase 1	−0.5544	0.7891
103511057	elongation of very long chain fatty acids protein 7	−0.3002	−0.1706
103513505	elongation of very long chain fatty acids protein 6	−0.3056	0.1593
103514113	fatty acid 2-hydroxylase%2C transcript variant X2	0.2158	0.2520
103507161	lipase 3-like	0.2464	−0.0516
103517998	elongation of very long chain fatty acids protein 1	0.7975	−0.5575
103520565	lipase 1-like	0.5060	−0.5310
103522648	long-chain fatty acid transport protein 4	−0.2287	0.0823
103512608	long-chain-fatty-acid-CoA ligase 6	−0.1295	0.3294
103516483	long-chain-fatty-acid-CoA ligase 4	−0.0160	0.2755

## Data Availability

Not applicable.

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
