# Peer review of "Silencing of Glycogen Synthase Kinase 3 Significantly Inhibits Chitin and Fatty Acid Metabolism in Asian Citrus Psyllid, Diaphorina citri"

_ijms, 2022, doi:10.3390/ijms23179654_

Round 1
Reviewer 1 Report
This MS entitled “Silencing of Glycogen Synthase Kinase 3 Significantly Inhibit Chitin and Fatty Acid Metabolism in Asian Citrus Psyllid, Diaphorina citri” by Zhang and colleagues describes the effect of Glycogen Synthase Kinase 3 (GSK3) on the adult ecdysis of D. citri. The results of GSK3 RNAi were clearly showed inhibitory effect on adult forming. GSK3 RNAi induced decrease of sugars and fatty acid. The authors also tried to clarify GSK3 signaling by RNA sequence and Q-PCR. These data suggest that chitin synthesis disorder is one cause of the disruption on adult ecdysis by GSK3 RNAi. The authors concluded that GSK was one of new targets for pest control.
The experiments have been designed well and the conclusions drawn are supported by data included in the MS. I have a few minor suggested change prior to publication.
1. The author mentioned that microinjection is a useful method for pest control. But it is very difficult to inject dsRNA into individual in field. Please reconsider about that.
2. The indexes of all figures are too small.
3. In the figure 3, statically differenced data were showed by different letters, a-d. But “d” was not found in the graphs.
4. The photos of GSK3 RNAi individuals were showed in figure 4c. It is better to put indicators for distinction some phenotypes in photos. Please indicate malformed type and abnormal tergites type.
5. Please mention why the cumulate molting rate of ds GFP injected were low.
6. Please mention the information of bar colors in Figure 6.
7. In section 2.5, please mention about UAP and CHT.
8. In section 2.6, “They were downregulated at 24 h or 48 h after silencing DcGSK3, including medium-chain-specific acyl-CoA dehydrogenase (MCAD), glutaryl-CoA dehydrogenase (GCD) and Lipase.” , is this correct? Please check.
9. The styles of supplemental table 2-4 are not suitable.
10. In M&M, please write in italics for “Murraya exotica”.
Author Response
- The author mentioned that microinjection is a useful method for pest control. But it is very difficult to inject dsRNA into individual in field. Please reconsider about that.
Reply: Thanks for your thoughtful and valuable comments. We also considered that microinjection of dsRNA is not very practical for pest control in the field. RNAi is a useful tool for gene functional research. Now, RNAi-based pest management strategies are now being employed for insect. However, many factors influence the efficiency of RNAi, especially for dsRNA-delivery method. There are three approaches to deliver dsRNA to insects, including oral feeding, microinjection and topical application. Oral feeding and topical application for dsRNA delivery can be used well in the field, while microinjection of dsRNA can only be used for fundamental research in the lab. We have added the related descriptions in previous manuscript, seeing Line 70 to Line 71.
- The indexes of all figures are too small.
Reply: Thanks for your valuable comments. We have revised the related figures in previous manuscript.
- In the figure 3, statically differenced data were showed by different letters, a-d. But “d” was not found in the graphs.
Reply: Thanks for your thoughtful comments. We have revised the incorrect descriptions in previous manuscript, seeing Line 147.
- The photos of GSK3 RNAi individuals were showed in figure 4c. It is better to put indicators for distinction some phenotypes in photos. Please indicate malformed type and abnormal tergites type.
Reply: Thanks for your valuable comments. We have revised the Figure 4 and added to previous manuscript, seeing Line 174.
- Please mention why the cumulate molting rate of dsGFP injected were low.
Reply: Thanks for your thoughtful and valuable comments. We found that the cumulate molting rate in the dsGFP groups was higher than the dsDcGSK3 groups (Figure 5C). In this study, RNAi was performed using fifth-instar D. citri nymphs. The fifth-instar nymphs must undergo normal molting before they can emerge into adults. Therefore, the cumulate molting rate is an important indicator. The results showed that the cumulative molting rate was increased from 12 h to 48 h after injection of dsGFP and dsDcGSK3, and the cumulative molting rate in the dsGFP groups is higher than the dsDcGSK3 groups. Chitin biosynthesis and degradation is one of the most important physiological processes during insect molting. We speculated that silencing of DcGSK3 decreased the expression levels of chitin-metabolism related genes, thus reduced the chitin content in D. citri. Finally, the D. citri cumulate molting rate was significantly reduced.
- Please mention the information of bar colors in Figure 6.
Reply: Thanks for your thoughtful comments. We have revised the Figure 6 and added to previous manuscript, seeing Line 203.
- In section 2.5, please mention about UAP and CHT.
Reply: Thanks for your valuable comments. We have added the relevant descriptions in previous manuscript, seeing Line 221 to Line 224.
- In section 2.6, “They were downregulated at 24 h and 48 h after silencing DcGSK3, including medium-chain-specific acyl-CoA dehydrogenase (MCAD), glutaryl-CoA dehydrogenase (GCD) and Lipase”, is this correct? Please check.
Reply: We are sorry for incorrect descriptions in previous manuscript. We have revised the related descriptions, seeing Line 242.
- The styles of supplemental table 2-4 are not suitable.
Reply: Thanks for your thoughtful comments. We have revised the styles of supplemental table 2-4.
- In M&M, please write in italics for “Murraya exotica”.
Reply: Thanks for your valuable comments. We have revised the species names to italics, seeing Line 454.
Reviewer 2 Report
To authors
Focus
The manuscript entitled " Silencing of Glycogen Synthase Kinase 3 Significantly Inhibit Chitin and Fatty Acid Metabolism in Asian Citrus Psyllid, Diaphorina citri" describes the silencing of DcGSK3 and the effects on Asian citrus psylid. The idea is good and the analysis mentioned supports the effects. Considering that this study may contribute to an improvement in the knowledge of the development of control Asian Citrus Psyllid, and IPM. The manuscript contributes to the literature and deserves publication.
I suggest to accept after minor revision the manuscript.
-Keywords should be in alphabetic order. Also, keywords serve to widen the opportunity to be retrieved from a database. To put words that already are into title and abstracts makes KW not useful. Please choose terms that are neither in the title nor in abstract.
-Change at first paragraph of introduction: mainly depents…insecticides by relies on the use of chemical insecticides
- Fig 3. Delete ‘Statistical analysis was conducted using SPSS software’ that should add at statistical analysis section at M&M, (the authors should do that to all figures)
Author Response
- Keywords should be in alphabetic order. Also, keywords serve to widen the opportunity to be retrieved from a database. To put words that already are into title and abstracts makes KW not useful. Please choose terms that are neither in the title nor in abstract.
Reply: Thanks for your valuable and thoughtful comments. We have revised the Keywords in previous manuscript, seeing Line 26 to Line 27.
- Change at first paragraph of introduction: mainly depents…insecticides by relies on the use of chemical insecticides
Reply: Thanks for your thoughtful comments. We have revised the related descriptions in previous manuscript, seeing Line 37 to Line 38.
- Delete ‘Statistical analysis was conducted using SPSS software’ that should add at statistical analysis section at M&M, (the authors should do that to all figures)
Reply: Thanks for your thoughtful suggestions. We have added a statistical analysis section at Materials and Methods in previous manuscript, seeing Line 574 to Line 577.